# Effects of Bacterial Metabolites on the Wnt4 Protein in Dental-Pulp-Stem-Cells-Based Endodontic Pulpitis Treatment

**DOI:** 10.3390/microorganisms11071764

**Published:** 2023-07-06

**Authors:** Ayman M. Abulhamael, Shilpa Bhandi, Nasreen H. Albar, Amal S. Shaiban, Shashit Shetty Bavabeedu, Khalid J. Alzahrani, Fuad M. Alzahrani, Ibrahim F. Halawani, Shankargouda Patil

**Affiliations:** 1Department of Endodontic, Faculty of Dentistry, King Abdulaziz University, Jeddah 21589, Saudi Arabia; amahmad4@kau.edu.sa; 2College of Dental Medicine, Roseman University of Health Sciences, South Jordan, UT 84095, USA; sbhandi@roseman.edu; 3Department of Restorative Dentistry, College of Dentistry, Jazan University, Jazan 45142, Saudi Arabia; nalbar01@gmail.com; 4Department of Restorative Dental Sciences, College of Dentistry, King Khalid University, Abha 62529, Saudi Arabia; ashiban@kku.edu.sa (A.S.S.); sbavabeedu@kku.edu.sa (S.S.B.); 5Department of Clinical Laboratories Sciences, College of Applied Medical Sciences, Taif University, Taif 21944, Saudi Arabia; ak.jamaan@tu.edu.sa (K.J.A.); fuadmubarak@tu.edu.sa (F.M.A.); i.halawani@tu.edu.sa (I.F.H.)

**Keywords:** dental pulp stem cell, docking, endodontic pulpitis, molecular simulation, *Porphyromonas gingivalis*, Wnt4

## Abstract

*Porphyromonas gingivalis* is associated with endodontic pulpitis, causing damage to the dental pulp, leading to severe pain and a decline in quality of life. Regenerative pulp treatments using dental pulp stem cells (DPSCs) can be hindered by interactions between DPSCs and the infecting bacteria. The protein WNT family member 4 (Wnt4) plays a critical role in the differentiation of DPSCs and the regeneration of odontogenic tissue. However, the specific influence of *P. gingivalis* on Wnt4 remains unclear. In this study, we employed a computational approach to investigate the underlying mechanisms through which *P. gingivalis*-produced metabolites inhibit the Wnt4 protein, thereby diminishing the regenerative potential and therapeutic efficacy of odontogenic tissue. Among the metabolites examined, C_29_H_46_N_7_O_18_P_3_S^−4^ exhibited the strongest inhibitory effect on the Wnt4 protein, as evidenced by the lowest binding energy score of −6782 kcal/mol. Molecular dynamic simulation trajectories revealed that the binding of C_29_H_46_N_7_O_18_P_3_S^−4^ significantly altered the structural dynamics and stability of the Wnt4 protein. These alterations in protein trajectories may have implications for the molecular function of Wnt4 and its associated pathways. Overall, our findings shed light on the inhibitory impact of *P. gingivalis*-produced metabolites on the Wnt4 protein. Further in vitro, in vivo, and clinical studies are necessary to validate and expand upon our findings.

## 1. Introduction

Untreated dental caries is a prevalent oral health issue affecting a significant proportion of the global population. According to the latest estimates from the Global Burden of Disease study, approximately 2.3 billion individuals suffer from untreated caries [1]. This represents a substantial public health concern with far-reaching implications as untreated dental caries can lead to oral health complications and systemic health issues. Irreversible pulpitis is one of the possible consequences of untreated dental caries and is characterised by inflammation of the dental pulp that cannot be reversed through therapeutic intervention. The inflammation is caused by bacterial infection, trauma, or extensive decay, resulting in irreversible damage to the pulp tissue. Symptoms of severe pain, often triggered by thermal or electrical stimuli, and spontaneous pain reduce the health-related quality of life [2]. The dominant genera linked to endodontic diseases include *Fusobacterium*, *Porphyromonas*, *Prevotella*, and others, which are often found in infected root canals. Bacteria found in the advanced layers of dentinal caries may serve as potential pathogens for inducing pulpitis and inciting pulp inflammation. *Porphyromonas gingivalis* is a Gram-negative anaerobic bacterium that is commonly found in the oral cavity, particularly in periodontal pockets and dental plaque. *P. gingivalis* has been found in infected root canals and has been detected at higher levels in teeth with irreversible pulpitis compared to teeth with reversible pulpitis [3,4]. Research has shown that *P. gingivalis* can invade and colonise the pulp chamber, which can induce inflammation and cause irreversible damage to the pulp tissue [5]. Treatment of pulpal infections in young permanent teeth in children and adolescents is a significant challenge for dental clinicians. Despite the availability of various treatment options, evidence-based treatment methods are limited to endodontic therapy, also known as root canal treatment, or extraction of the affected tooth. Endodontic therapy is fraught with challenges, such as complexity and variability in root canal anatomy, microbial challenges, handling complications, and addressing patient-related factors that significantly impact treatment outcomes and necessitate careful consideration and expertise. Recently, stem-cell-based regeneration of odontogenic tissues has gained popularity due to its non-invasive nature [6,7].

Dental pulp stem cells (DPSCs) are a type of mesenchymal stem cell found within the dental pulp of teeth [8,9]. They have the ability to differentiate into various cell types, including odontoblasts, osteoblasts, adipocytes, and chondrocytes [10,11,12]. DPSCs have been investigated as a potential alternative treatment option for pulpitis due to their ability to differentiate into odontoblasts, the cells responsible for the formation of dentin, the hard tissue that surrounds the pulp chamber, and root canals [13]. DPSCs possess strong angiogenic potential, which enables them to generate capillary-like structures by releasing angiogenic regulators [14]. This feature makes DPSCs a valuable resource for endodontic treatment, particularly for root canal therapy, where infected pulp can be replaced with scaffolds containing DPSCs and signalling molecules to aid in differentiation [15]. However, in an infectious state, several inflammatory factors alter the natural microenvironment of DPSCs and hinder odontogenic development by dysregulating the signalling molecules and their associated pathways [16]. The dysregulation of signalling molecules refers to the disturbance or abnormality in the normal patterns of molecular signalling pathways that are involved in regulating the growth, differentiation, and maturation of DPSCs. These pathways include important regulators of odontogenic development, such as the Wnt signalling pathway, the BMP signalling pathway, and the Notch signalling pathway. The disruption of these signalling molecules and pathways in the infectious state hampers the proper functioning and differentiation of DPSCs, thereby hindering odontogenic development and resulting in impaired tooth formation and compromised tissue regeneration. Understanding the mechanisms by which inflammatory factors dysregulate signalling molecules and associated pathways is crucial for developing strategies to mitigate the negative impact of infections on dental pulp tissue and promote successful tooth development and regeneration.

The Wnt4 protein, also known as Wingless-related Integration Site 4, belongs to the Wnt family of secreted glycoproteins. Wnt proteins play crucial roles in various biological processes, including embryonic development, tissue homeostasis, and cell fate determination [17,18]. Wnt4 is specifically involved in signalling pathways that regulate cell proliferation, differentiation, and tissue morphogenesis. The Wnt4 protein plays a role in both normal physiological conditions and pathological conditions [19]. Research conducted by Zhong et al. demonstrated the regulatory role of Wnt4 in dental pulp stem cells under inflammatory conditions. Notably, DPSCs from normal dental pulp, when exposed to the inflammatory microenvironment, showed decreased expression of Wnt4, which correlates significantly with the decreased odontogenic potential of DPSCs [20]. The decrease in Wnt4 expression suggests that the inflammatory conditions negatively impact the regulatory mechanisms involved in maintaining the odontogenic potential of DPSCs, implying a disturbance in the normal signalling network required for proper odontogenesis. Understanding the mechanism of *Porphyromonas gingivalis*-induced inflammation and tissue destruction in the dental pulp and its relationship with the Wnt4 signalling pathway can help to identify potential therapeutic targets. Additionally, understanding the effects of bacterial metabolites on the Wnt4 protein, which plays a crucial role in cellular signalling and tissue homeostasis, can provide valuable insights into the pathogenesis of endodontic pulpitis and contribute to the development of effective treatment strategies. Investigating the interaction between bacterial metabolites and the Wnt4 protein in dental pulp stem cells can shed light on the underlying mechanisms and potentially identify novel therapeutic targets for endodontic pulpitis. This research can help to advance the knowledge base on using pluripotent stem cells for the regenerative treatment of *P. gingivalis*-induced pulpitis. The current investigation aimed to determine the inhibitory effects of secreted metabolites produced by *Porphyromonas gingivalis* on human Wnt4 protein through a computational approach.

## 2. Materials and Methods

We conducted a sequential investigation in order to evaluate the inhibitory potential of secreted metabolites produced by *Porphyromonas gingivalis* on human Wnt4 protein. Our methodology included the following steps: (1) generation of a structure for the Wnt4 protein using a homology-based approach; (2) retrieval, optimisation, and molecular docking of *Porphyromonas gingivalis*-secreted metabolites with the Wnt4 protein; (3) molecular dynamics simulation to analyse the structural dynamics, stability, and binding affinity of the selected *Porphyromonas gingivalis*-secreted metabolite and its potential impact on stem cell odontogenic differentiation through the Wnt signalling pathway.

### 2.1. Retrieval of Protein Sequence

The protein sequence of Wnt family member 4 (Wnt4) with the accession number P56705 (WNT4_HUMAN) was retrieved from the UniProt database. The sequence was subjected to the NCBI-Blastp tool (https://blast.ncbi.nlm.nih.gov/Blast.cgi (accessed on 18 February 2023) to look for any available human three-dimensional structure of the Wnt4 protein in the PDB database (https://www.rcsb.org/ (accessed on 20 February 2023). Due to a lack of structure in the PDB database, the sequence of the Wnt4 protein was used to search for the most similar protein with the experimentally elucidated structures. These collected protein structures were assessed based on percentage identity, E-value, query coverage, and alignment score for use as a template for the generation of the three-dimensional Wnt4 protein structure. From the selected template using MODELLER version 9.15, the Wnt4 structure was generated. For the given template, MODELLER provided several modelled structures. The DOPE score was used to select the best-modelled Wnt4 structure for quality assessment [21].

### 2.2. Validation of the Structure

The SAVES (https://saves.mbi.ucla.edu/ (accessed on 2 March 2023) server was used to validate the quality of the modelled Wnt4 protein structure with QMEAN, VERIFY3D, and PROCHECK. The PROCHECK assesses the stereochemical quality using the phi/psi angle arrangement in the Ramachandran plot. Similarly, VERIFY3D was used to calculate a compatibility value for each protein residue and provide the quantity score for the structure. In the same way, QMEAN evaluates the quality of a structure based on its physical and chemical properties. It then assigns the structure an overall quality score and compares it to the estimated QMEAN-scores of high-resolution experimental structures [22].

### 2.3. Retrieval and Preparation of Microbial Metabolites

The virtual metabolic human database (VMH) (https://www.vmh.life/ (accessed on 12 March 2023)) provides human microbiome resources containing metabolite information on 605 species. We collected the *Porphyromonas gingivalis*-secreted metabolites in the given SMILES (Simplified Molecular Input Line Entry System) format from VMH. The Open Babble version 2.4.1 software was used to convert the SMILES into accessible Structure Data File (SDF) format for molecular docking.

### 2.4. Molecular Docking

The preparation of the Wnt4 protein and microbial metabolites for molecular docking was performed using Maestro 11.2 version, Schrodinger suite. Before molecular docking, all the SDF metabolite structures were energy-minimised with the OPLS4 force field. Next, the Wnt4 protein structure was pre-processed with the protein preparation wizard, and then the grid box was generated around the active site (PHE^238^, ASP^239^, SER^276^, PRO^277^, PHE^279^, ILE^303^, ARG^85^, ARG^86^, TRP^87^, ARG^107^) identified using the P2RANK server (http://siret.ms.mff.cuni.cz/p2rank (accessed on 18 March 2023). The generated grid box using a receptor grid generation module provided the confined location in the protein for the metabolite’s binding. Glide docking was performed, and the metabolites of *Porphyromonas gingivalis* inhibiting Wnt4 were screened based on their least binding energy (kcal/mol).

### 2.5. Molecular Dynamics Simulations (MDS)

Based on the outcome of molecular docking, molecular dynamics simulations were performed. The microbial metabolite that showed the least binding energy with Wnt4 was compared with the metabolite-free Wnt4 for the MDS study. The molecular dynamics simulation was run using the GROMACS 2021 package. A CHARMM-all atoms force field was applied, and an orthorhombic periodic boundary box with a dimension of 1 Å was generated for the metabolite–WNT4 complex and free WNT, respectively. A TIP3P water model was used to solvate, and ions were added to neutralise the system. A steep descent strategy of 50,000 steps was applied, and the system was equilibrated based on NVT (number of particles, volume, and temperature) at 300 K and NPT (number of particles, pressure, and temperature) at 1 bar of pressure. The production MD run was carried out for 100 ns by implementing Particle Mesh Ewald (PME). The MD trajectories were plotted using Microsoft Excel (Microsoft Inc., Redwood, CA, USA) to show the differences between complex and free WNT based on root mean square deviation (RMSD), root mean square fluctuation (RMSF), radius of gyration (RG), and solvent-accessible surface area (SASA).

## 3. Results

### 3.1. WNT4 Protein Sequence and Structure Generation

The human Wnt4 protein sequence was retrieved from the Uniprot database using accession number P56705, which contains 351 amino acids. The NCBI Blastp against the PDB database showed the closest structural template (PDB accession 7DRT) based on the assessment, including e-value, percentage identity, alignment score, and query coverage. Notably, the 7DRT sequence showed 2e^−118^ e-value, 49.25% identity, 347 as an alignment score, and 93% coverage with the Wnt4. Thereby, 7DRT was selected as the best template to construct a 3D structure for Wnt4 using MODELLER software version 9.11. Among the several modelled structures for Wnt4, the best structure was selected to have −321 as a DOPE score. The modelled structure was assessed using PROCHECK, Verify3D, and QMEAN. The PROCHECK based on the Ramachandran plot showed that 83.8% of residues were in favourable regions, confirming that the generated model was of high quality. Similarly, Verify3D confirmed that 82.6% of the Wnt4 residues in the structure were of high quality. The QMEAN was observed to be 0.97, which demonstrated the high resolution of the modelled structure and was almost the same as the experimental structures. These results demonstrate the successful construction of a reliable 3D model for the Wnt4 protein using the selected template. The assessments conducted through PROCHECK, Verify3D, and QMEAN provide confidence in the accuracy and quality of the model.

### 3.2. Porphyromonas gingivalis—Derived Metabolites and Molecular Docking

The VMH database was used to collect the metabolites produced by *P. gingivalis* in the SMILES format. Overall, 870 metabolites of *P. gingivalis* were collected and converted into SDF format for molecular docking. All the analysed metabolites showed a wide range of binding energy between −6.782 and 2.064 kcal/mol. Figure 1 shows the top ten microbial metabolites and their binding energies. Of these, C_29_H_46_N_7_O_18_P_3_S^−4^(Pubchem ID: 46173319) was observed to have the least binding energy (−6.782 kcal/mol) at the active site of the Wnt4 protein (Figure 2). Based on the ligand interaction diagram, C_29_H_46_N_7_O_18_P_3_S^−4^ forms hydrogen bond interactions with Wnt4 at GLN^81^, ARG^85^, ARG^86^, TRP^87^, ARG^107^, ASP^239^, and PRO^277^. These crucial interactions may cause a change in the structure of Wnt4, leading to changes in the behaviour and function of the protein. These findings highlight the significant binding affinity of C_29_H_46_N_7_O_18_P_3_S^−4^ towards the active site of Wnt4. The identified hydrogen bond interactions with specific amino acid residues suggest a potential mechanism by which C_29_H_46_N_7_O_18_P_3_S^−4^ may modulate the structure and function of Wnt4.

### 3.3. Dynamic Simulation of WNT4 with a Microbial Inhibitor

MDS was run for up to 100 nanoseconds to investigate the dynamic behaviour of the Wnt4–C_29_H_46_N_7_O_18_P_3_S^−4^ complex and compare it with the metabolite-free Wnt4 protein. The comparative trajectories of RMSD, RG, SASA, and RMSF are displayed in Figure 3, Figure 4, Figure 5 and Figure 6. All the analysed trajectories showed differences between the complex and free Wnt4 proteins. The RMSD plot showed fluctuation at 0 and 100 ns, with an average change in RMSD of 0.55 nm for free and 0.974 nm for the complex on C_29_H_46_N_7_O_18_P_3_S^−4^ binding (Figure 3). Similarly, the RMSF plot also revealed fluctuations throughout the amino acid residues on the C_29_H_46_N_7_O_18_P_3_S^−4^ interaction (Figure 4). The residues at the active sites GLN^81^, ARG^85^, ARG^86^, TRP^87^, ARG^107^, ASP^239^, and PRO^277^ were observed to have increased fluctuations. The SASA and RG plots have been used to illustrate the distinctions between the free and Wnt4 complex states. The mean SASA (as depicted in Figure 5) and RG (as depicted in Figure 6) for the free state were 190.96 nm^2^ and 2.33 nm, respectively. In contrast, the mean SASA and RG for the complex state based on the simulation were higher, i.e., 192.10 nm^2^ and 2.66 nm, respectively. These results indicate that the binding of C_29_H_46_N_7_O_18_P_3_S^−4^ to Wnt4 influences the dynamic behaviour and conformational changes of the protein. The observed fluctuations in RMSD and RMSF suggest that the complex experiences structural rearrangements compared to the free Wnt4 protein. Additionally, the alterations in SASA and RG values further support the notion that the complex state exhibits distinct characteristics from the free state.

## 4. Discussion

Dental pulp stem cells (DPSCs), found within the dental pulp of teeth, have the potential to differentiate into different cell types that are able to contribute to the repair and regeneration of damaged or infected dental pulp tissue, which can help to reduce inflammation and promote healing of pulpitis, making them a promising alternative treatment option [23]. The Wnt4 signalling pathway is involved in the differentiation of dental pulp cells and odontoblasts, the cells that form dentin. Wnt4 signalling is important in the process of tooth repair and regeneration. Alterations in the Wnt4 signalling pathway are associated with various dental diseases, such as pulpitis, and further research on this signalling pathway could lead to new therapies for treating these conditions. In order to avoid the risk of immunological rejection, it is recommended to collect DPSCs from the same patient who is undergoing the endodontic procedure. However, the presence of *Porphyromonas gingivalis*, which is commonly found in irreversible pulpitis patients, may affect the efficacy of the treatment if it is not effectively eliminated [24,25]. In the present study, we aimed to evaluate the inhibitory potential of secreted metabolites produced by *P. gingivalis* on human Wnt4 protein using a computational approach.

Our computational approach demonstrated that the microbial metabolite has an effect on the Wnt4 protein, which is a key regulator of stem cell properties. We established the crucial top ten metabolites secreted by *P. gingivalis* that showed a significant binding affinity with Wnt4 protein (Figure 1). Based on these results, C_29_H_46_N_7_O_18_P_3_S^−4^ from *P. gingivalis* effectively inhibits the Wnt4 protein. Through the docking protocol, C_29_H_46_N_7_O_18_P_3_S^−4^ was fitted inside the binding pocket of the Wnt4 protein, showing the interaction with ARG^107^, GLN^81^, ARG^85^, ARG^86^, TRP^87^, ARG^107^, ASP^239^, and PRO^277^ (binding energy = −6.782 kcal/mol). Interestingly, C_29_H_46_N_7_O_18_P_3_S^−4^ forms a greater number of hydrogen bonds with the Wnt4 protein, which is crucial to have strong inhibitory activity (Figure 2). Despite limitations in the ability of molecular docking methods to account for protein molecule flexibility within a cellular environment, we employed a modified approach, MDS, to analyse both free Wnt4 and Wnt4 in complex with C_29_H_46_N_7_O_18_P_3_S^−4^ in order to circumvent these constraints. The RMSD (root mean square deviation) is a measure of the deviation of the atoms in a structure from their positions in a reference structure. In order to gain insight into the interaction between Wnt4 and C_29_H_46_N_7_O_18_P_3_S^−4^, molecular dynamic simulations were conducted on the docked complex.

The stability of the complex was evaluated by analysing the RMSD trajectories. A low average RMSD value indicates high stability [26,27]. The average RMSD for free Wnt4 was found to be 0.55 nm, while the average RMSD for the Wnt4 complex was 0.974 nm. The lower the RMSD value, the closer the atoms are to their positions in the reference structure. In this case, the RMSD of 0.974 nm for the Wnt4 complex implies that the complex is less stable than the free Wnt4 molecule as it is farther from the reference structure. This can mean that the interaction between Wnt4 and C_29_H_46_N_7_O_18_P_3_S^−4^ is not as strong as it could be or that the complex is less stable in the simulated environment. The RMSD plot for the Wnt4–C_29_H_46_N_7_O_18_P_3_S^−4^ complex (Figure 3) revealed high fluctuations compared to the free Wnt4, indicating a loss of stability upon complex formation. The RG trajectories were utilised to evaluate the structural compactness of protein complexes that contained both homogenous and heterogeneous components [28]. Smaller RG values indicate greater stability, while larger values indicate instability [29]. The correlation between the RMSD and RG plots for the Wnt4 complex suggests that the binding of C_29_H_46_N_7_O_18_P_3_S^−4^ alters the structural stability of the Wnt4 protein. This implies that the binding of C_29_H_46_N_7_O_18_P_3_S^−4^ to Wnt4 causes a change in the stability of the Wnt4 protein. The RMSD plot indicates that the atoms in the Wnt4–C_29_H_46_N_7_O_18_P_3_S^−4^ complex deviate more from their positions in the reference structure (free Wnt4) than when Wnt4 is unbound, indicating that the complex is less stable than the free Wnt4 molecule. As shown in Figure 6, there was a notable difference in the RG values between free Wnt4 (2.33 nm) and the Wnt4 complex (2.64 nm). These results confirm the stability of the structure determined from the RMSD plot. The RG revealed that the Wnt4 protein with C_29_H_46_N_7_O_18_P_3_S^−4^ causes a decrease in structural compactness (2.66 nm) and stability when compared to free Wnt4. A larger RG value for the Wnt4–C_29_H_46_N_7_O_18_P_3_S^−4^ complex than for the free Wnt4 protein indicates that the complex is less compact and, thus, less stable than the free Wnt4 protein. Overall, these findings suggest that the binding of C_29_H_46_N_7_O_18_P_3_S^−4^ to Wnt4 alters the structural stability and compactness of the Wnt4 protein, potentially indicating a weaker interaction or reduced stability in the simulated environment. The deviation from the reference structure and the differences in RG values collectively indicate that the Wnt4 complex is less stable and less compact than the free Wnt4 protein, which could affect the overall function and behaviour of the Wnt4 protein. The decrease in structural compactness and stability indicated by the larger RG values further supports the notion that the complex is less compact and stable than the unbound Wnt4 protein.

Analysis of solvent-accessible surface area (SASA) and root mean square fluctuation (RMSF) plots provided insights into the impact of complex formation and drug binding on the structural stability and flexibility of the Wnt4 protein. SASA (solvent-accessible surface area) plots are used to measure the solvent exposure of a protein’s structure. Lower SASA values correspond to increased hydrophobic amino acid residues, which can contribute to increased system stability [30]. We observed marginal differences in the SASA plot of free Wnt4 and the Wnt4 complex, with SASA averages of 190.96 nm^2^ and 192.11 nm^2^, respectively (Figure 5). This implies that there is no significant change in the solvent exposure of the protein’s structure upon complex formation. The SASA values for the free Wnt4 and the complex are very close, which suggests that the complex and free Wnt4 have similar levels of solvent exposure. This may indicate that the interaction between Wnt4 and C_29_H_46_N_7_O_18_P_3_S^−4^ does not have a significant impact on the protein’s surface properties. It may also imply that the simulation conditions did not allow for a significant change in the solvent exposure of the protein. The observation of a marginal difference in the SASA plot may indicate a decrease in structural stability as the binding of C_29_H_46_N_7_O_18_P_3_S^−4^ leads to less exposure of Wnt4’s hydrophobic amino acid residues. The RMSF (root mean square fluctuation) can provide insight into the effect of drug binding on the Wnt4 amino acid residues, with larger RMSF values indicating increased flexibility of the alpha carbon atoms. The RMSF is a measure of the fluctuation of amino acid residues in Wnt4 as a result of drug binding, and an increase in the RMSF value suggests an increase in the flexible movements of the alpha carbon atoms. The binding of C_29_H_46_N_7_O_18_P_3_S^−4^ results in a marginal increase in the average RMSF value for Wnt4 (0.376 nm) compared to free Wnt4 (0.255 nm), as shown in Figure 4, indicating that the microbial metabolite (C_29_H_46_N_7_O_18_P_3_S^−4^) causes an overall increase in the flexibility of Wnt4’s amino acid residues. This increased flexibility in the amino acid residues of the Wnt4 complex is consistent with the observations from the RMSD, RG, and SASA plots. The implications of these findings are that the binding of C_29_H_46_N_7_O_18_P_3_S^−4^ to Wnt4 may slightly decrease its structural stability by reducing the exposure of hydrophobic amino acid residues. Additionally, it increases the flexibility of Wnt4’s amino acid residues. These changes in stability and flexibility could have implications for the function and behaviour of the Wnt4 protein.

In conclusion, our computational analysis has provided valuable insights into the identification of a pivotal metabolite that inhibits the beneficial Wnt4 protein, potentially impeding the regenerative capacity of odontogenic treatments. A pressing need in combating the spread of *P. gingivalis* is the development of novel strategies for controlling or treating recurrent infections. Within this context, our research has identified a critical factor associated with *P. gingivalis* that significantly influences treatment outcomes, warranting careful consideration in the development of new therapeutic approaches. As a result, this investigation lays the groundwork for future endeavours in the design and synthesis of novel agents aimed at safeguarding DPSCs by targeting the Wnt4 protein. The identified interaction between DPSCs and C_29_H_46_N_7_O_18_P_3_S^−4^, as revealed by our computational methods, necessitates experimental evaluation, serving as a fundamental basis for advancement in innovative treatment modalities to enhance DPSC-based odontogenic-tissue-regenerative therapies.

## 5. Conclusions

Using DPSCs obtained from individuals with periodontitis for endodontic regenerative procedures may impact the success of the treatment. Our computational study has identified a crucial metabolite produced by *Porphyromonas gingivalis* predicted to inhibit and destabilise the Wnt4 protein of human dental pulp stem cells. The binding of this microbial metabolite may lead to modifications in Wnt4 that could impede the odontogenic differentiation capacity of DPSCs. Further experiments are needed to investigate the effects of this microbial metabolite on stem cell therapy.

## Figures and Tables

**Figure 1 microorganisms-11-01764-f001:**
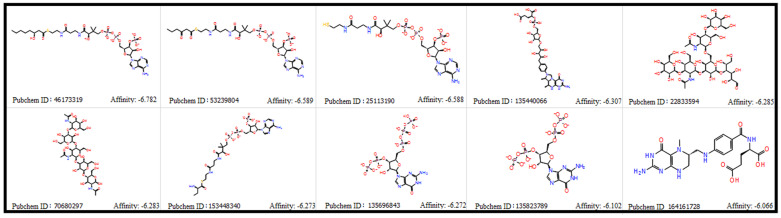
Top ten microbial metabolites, showing binding energies ranging from −6.78 to −6.066 kcal/mol against the Wnt4 protein.

**Figure 2 microorganisms-11-01764-f002:**
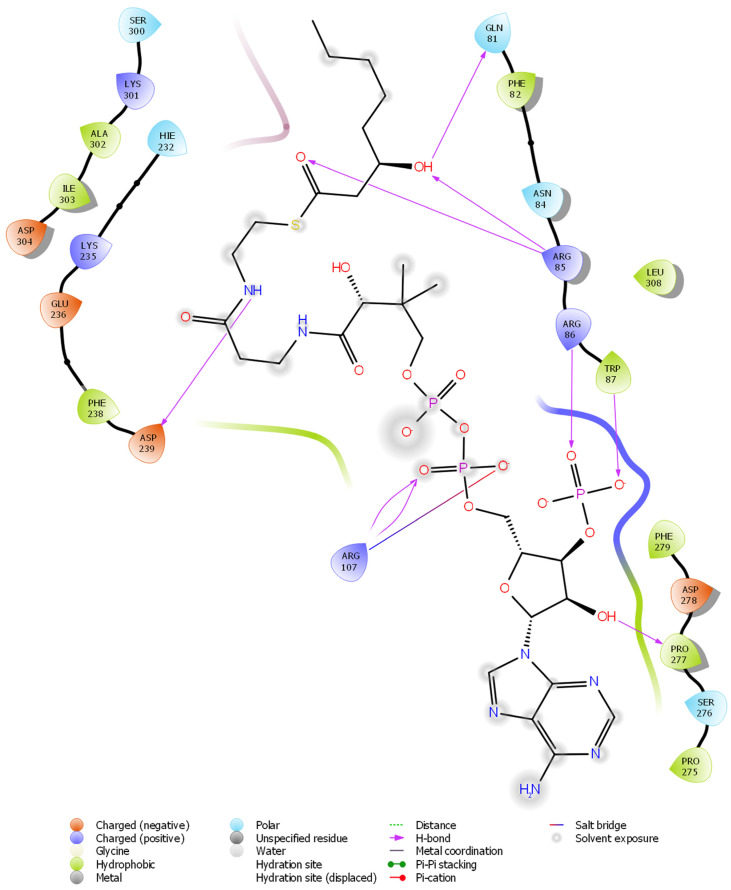
C_29_H_46_N_7_O_18_P_3_S^−4^ of *Porphyromonas gingivalis* interacting with the crucial amino acid residues of Wnt4.

**Figure 3 microorganisms-11-01764-f003:**
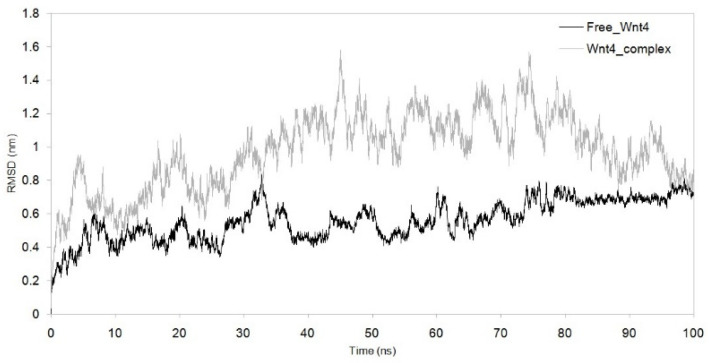
RMSD showing difference in trajectories with the function of time (100 ns) between complex and free Wnt4 protein. Black: free Wnt4 protein; grey:Wnt4 protein with C_29_H_46_N_7_O_18_P_3_S^−4^ complex.

**Figure 4 microorganisms-11-01764-f004:**
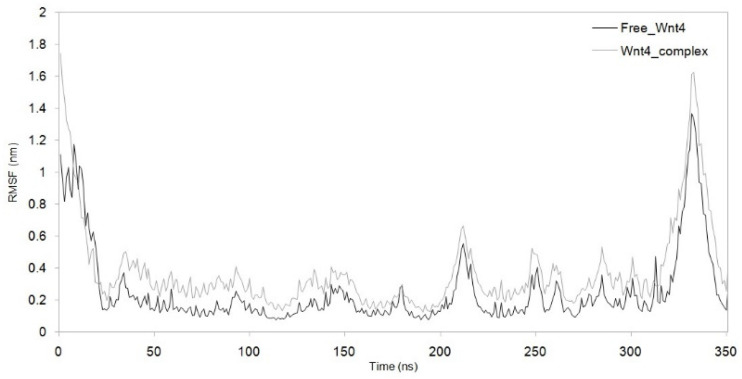
RMSF showing difference in residues fluctuation between complex and free Wnt4 protein. Black: free Wnt4 protein; grey:Wnt4 protein with C_29_H_46_N_7_O_18_P_3_S^−4^ complex.

**Figure 5 microorganisms-11-01764-f005:**
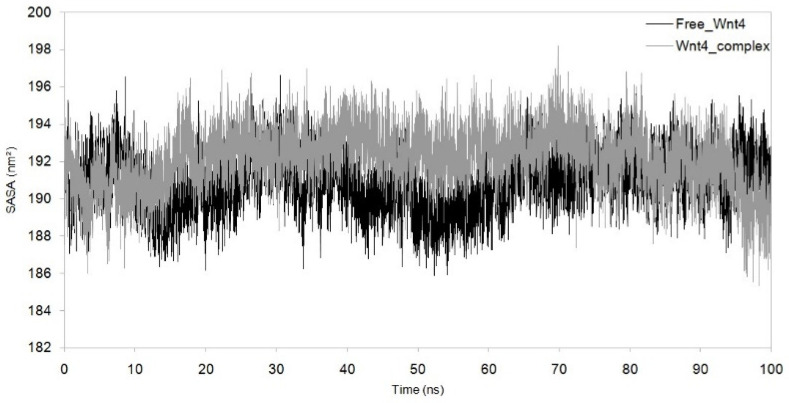
Solvent-accessible surface area (SASA) with function of time obtained for Wnt4–C_29_H_46_N_7_O_18_P_3_S^−4^ complex with reference to free Wnt4. Black: free Wnt4 protein; grey:Wnt4 protein with C_29_H_46_N_7_O_18_P_3_S^−4^ complex.

**Figure 6 microorganisms-11-01764-f006:**
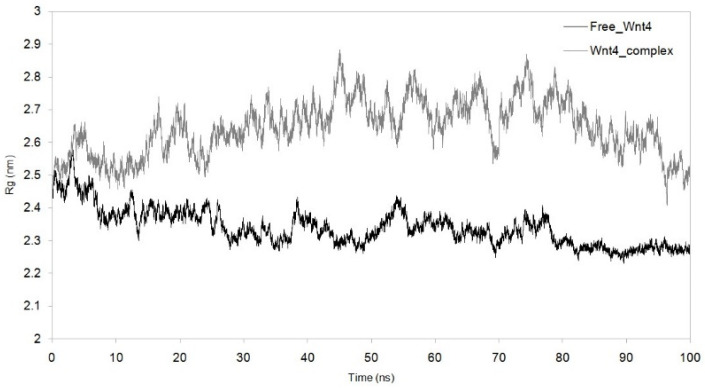
Radius of gyration (Rg) with function of time obtained for Wnt4–C_29_H_46_N_7_O_18_P_3_S^−4^ complex with reference to free Wnt4. Black: free Wnt4 protein;grey: Wnt4 protein with C_29_H_46_N_7_O_18_P_3_S^−4^ complex.

## Data Availability

Not applicable.

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
