# Peer review of "Effects of Bacterial Metabolites on the Wnt4 Protein in Dental-Pulp-Stem-Cells-Based Endodontic Pulpitis Treatment"

_microorganisms, 2023, doi:10.3390/microorganisms11071764_

Round 1

Reviewer 1 Report

Although the study is valuable, it has some shortcomings.

Various situations should be considered that will increase the research value.

Abstract should be rewritten with clear objectives and scientific language

Add the significance of your work.

Add limitation of the study

The typos should be corrected. The article should be accepted after Minor revision.

Author Response

Reviewer 1:

Comment1: Although the study is valuable, it has some shortcomings.

Response: Thank you for taking the time to review our study. We appreciate your feedback and value your insights regarding the shortcomings of our research.

Comment2: Abstract should be rewritten with clear objectives and scientific language.  

Response: Thank you for your comment. We have revised the abstract in line with your recommendations.

Comment2: Add the significance of your work, Add limitation of the study.

Response: We sincerely appreciate your guidance in this matter. We have revised the discussion to include lines describing the significance and limitations of our study.

Reviewer 2 Report

The title of the paper is well formulated and it covers the content. The abstract contains the main required components, and it coherently covers the main research question developed in the manuscript by pointing out the logical conclusions and limitations of the paper. The introduction logically follows the aim of the paper and provides a valuable introspection into an unsolved topic. Methodological part of the paper is suits current scientific standards and results are presented and discussed clearly. The interpretation of tables and figures is acceptable. However, wider context of the presentation of the results should be applied to make the results really understandable for the audience. The level of the author’s knowledge is satisfying. It is obvious that authors are well oriented in the topic and that they use appropriate terms. The overall level of language is appropriate. The paper is suitable to be published after minor corrections  according to the comments included in the review:

- extend the discussion section, pointing out the potential application in the market, as well as provide data about potential costs of such innovations;

- The authors do not discuss possible limitations of their study or the insights for future directions of research. Maybe they could discuss external validity of the results in terms of possible insights in real context/large scale production and/or additional variables that they would have liked to have to better answer to their research question.

- I recommend that authors review the article thoroughly and consider using a professional proofreading service to improve the style of the article. Many sentences are unclear.

The language is satisfactory

Author Response

Reviewer 2:

Comment1: - extend the discussion section, pointing out the potential application in the market, as well as provide data about potential costs of such innovations.

Response: Thanks for the comment the suggested changes were incorporated in the revised manuscript.

Comment 2: - The authors do not discuss possible limitations of their study or the insights for future directions of research. Maybe they could discuss external validity of the results in terms of possible insights in real context/large scale production and/or additional variables that they would have liked to have to better answer to their research question.

Response: Thanks for the comment the suggested changes were incorporated in the revised manuscript.

Comment 3: - I recommend that authors review the article thoroughly and consider using a professional proofreading service to improve the style of the article. Many sentences are unclear.

Response: The manuscript is now revised extensively by an English language professional.
